# Purcell-enhanced quantum yield from carbon nanotube excitons coupled to plasmonic nanocavities

Yue Luo [1,2], Ehsaneh D. Ahmadi[1], Kamran Shayan[1,2], Yichen Ma[1,2], Kevin S. Mistry[3], Changjian Zhang[4], James Hone[4], Jeffrey L. Blackburn[3] & Stefan Strauf [1,2]

Single-walled carbon nanotubes (SWCNTs) are promising absorbers and emitters to enable novel photonic applications and devices but are also known to suffer from low optical quantum yields. Here we demonstrate SWCNT excitons coupled to plasmonic nanocavity arrays reaching deeply into the Purcell regime with Purcell factors ($F_P$) up to $F_P = 180$ (average $F_P = 57$), Purcell-enhanced quantum yields of 62% (average 42%), and a photon emission rate of 15 MHz into the first lens. The cavity coupling is quasi-deterministic since the photophysical properties of every SWCNT are enhanced by at least one order of magnitude. Furthermore, the measured ultra-narrow exciton linewidth (18 µeV) reaches the radiative lifetime limit, which is promising towards generation of transform-limited single photons. To demonstrate utility beyond quantum light sources we show that nanocavity-coupled SWCNTs perform as single-molecule thermometers detecting plasmonically induced heat at cryogenic temperatures in a unique interplay of excitons, phonons, and plasmons at the nanoscale.

[1] Department of Physics, Stevens Institute of Technology, Hoboken, NJ 07030, USA. [2] Center for Quantum Science and Engineering, Stevens Institute of Technology, Hoboken, NJ 07030, USA. [3] National Renewable Energy Laboratory, Golden, CO 80401, USA. [4] Department of Mechanical Engineering, Columbia University, New York, NY 10027, USA. Correspondence and requests for materials should be addressed to S.S. (email: strauf@stevens.edu)

Single-walled carbon nanotubes (SWCNTs) are promising absorbers and emitters in applications including solar energy conversion[1], biological imaging[2], and on-chip quantum photonics such as room-temperature single photon emission in oxygen-doped SWCNTs[3] and quantum optical circuits with integrated superconducting detectors[4]. Optical quantum yields from SWCNTs remain nevertheless low (2–7%)[5–9] and light enhancement strategies are required for practical implementations to improve quantum yield[10], preserve exciton coherence[11], and potentially enable indistinguishable photons[12]. Early attempts to couple SWCNT excitons to planar surface plasmons show however no Purcell effect[13], while dielectric cavities display moderate Purcell factors of $F_P = 5$ (ref. [7]).

A promising alternative to the dielectric cavity is the plasmonic nanocavity that features nanometer sized gaps with ultra-small mode volumes resulting in drastically enhanced spontaneous emission (SE) rates of quantum emitters[10,14–16]. Plasmonics can be lossy when dipole emitters are placed in close proximity to isolated metal nanoparticles leading to dominant nonradiative (NR) recombination, and thus photoluminescence (PL) quenching[17]. This, however, is not the case for nanoplasmonic resonators, which can feature a dominant radiative recombination rate with a branching ratio of 75% radiative to 25% NR when emitters are in close proximity (2 nm) to nanocavity gap-modes[18]. Importantly, the higher-order dark modes in isolated plasmonic nanoparticles become hybrid-plasmonic dipole modes in both bowtie nanoantennas and gap-mode structures, resulting in

massively enhanced emitter decay via radiative channels[19]. This effect recently led to the first experimental demonstration of a single molecule entering the strong-coupling regime with a plasmonic gap-mode[20]. Remarkable results have also been demonstrated for dye molecules coupled to elliptical gold nanoparticles with intensity enhancement factors (EF) of about 100 (ref. [21]), single molecule emission from antenna-in-a-box platforms (EF ~ 1100)[22], and bowtie nanoantennas (EF ~ 1340)[23]. Using plasmonic gap modes, large Purcell factors up to $F_P = 1000$ were also demonstrated[10,18]. Plasmonic nanocavities are thus an appealing route to overcome the low quantum yield of carbon nanotubes and to potentially enhance efficiencies in device applications.

Here we demonstrate SWCNTs excitons coupled to dense arrays of plasmonic nanocavities reaching deeply into the Purcell regime with $F_P = 180$, Purcell-enhanced quantum yields up to 62% from initially 2%, and a photon emission rate of 15 MHz into the first lens. The coupling is quasi-deterministic since the intensity and quantum yield of each of the 21 investigated SWCNTs located on the nanogap array is enhanced by at least one order of magnitude. The simultaneous observation of an ultra-narrow exciton linewidth (18 μeV, $T_2 = 73$ ps) and a Purcell enhanced lifetime (37 ps) is promising towards future generation of transform-limited photons from a SWCNT. In addition, we show that nanocavity-coupled SWCNTs perform as single-molecule thermometers detecting plasmonically induced heat in the temperature range from 4 to 120 K.

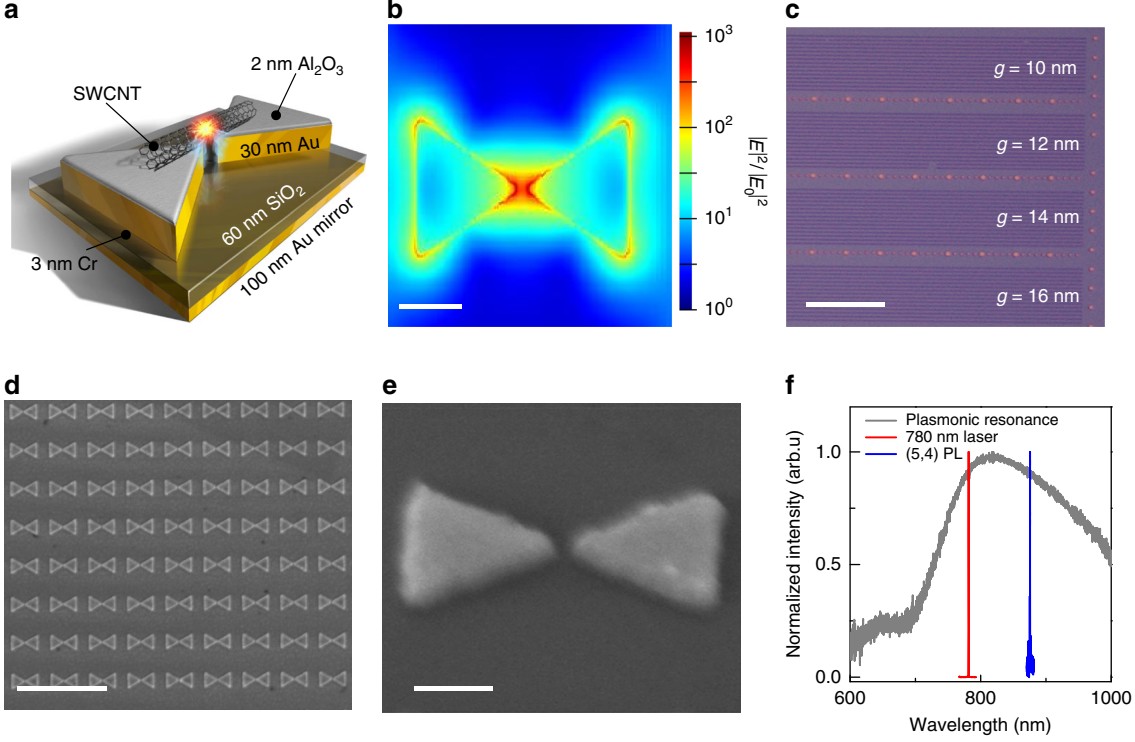

**Fig. 1** Overview of plasmonically coupled carbon nanotube system. **a** Shematic of an individual single-wall carbon nanotube (SWCNT) suspended across a bowtie antenna. The SWCNT ($d < 1$ nm) is portrayed with significantly larger scale than actual size for clarity. The SWCNT is separated from the plasmonic gold substrate by a 2 nm atomic layer deposition grown $Al_2O_3$ spacer layer to prevent optical quenching and short circuit of the nanoplasmonic gap-mode underneath. **b** Finite-difference time-domain (FDTD) simulation of the corresponding field enhancement distribution profile including finite apex angles with 3 nm radius. Scale bar=100 nm. **c** Bright-field optical microscope image of the plasmonic array showing four $20 \times 100$ μm$^2$ stripes each containing bowtie antennas with fixed gap size $g$ varying among stripes from 10–20 nm. The larger features are gold markers to enable repositioning to individual SWCNTs. Scale bar=20 μm. **d** The scanning electron microscope image shows high uniformity and orientation control of the plasmonic system. Scale bar=2 μm. **e** Zoom into an individual bowtie antenna with 10 nm gap showing sharp and straigth edges. Scale bar=100 nm. **f** Plasmon resonance spectrum ($Q = 6$) recorded in dark-field transmision geometry (gray) together with 780 nm pump laser spectrum (red) and exciton emission spectrum of a (5, 4) SWCNT (blue) showing spectral resonance is fullfilled simultaneously for both SWCNT absorption and emission

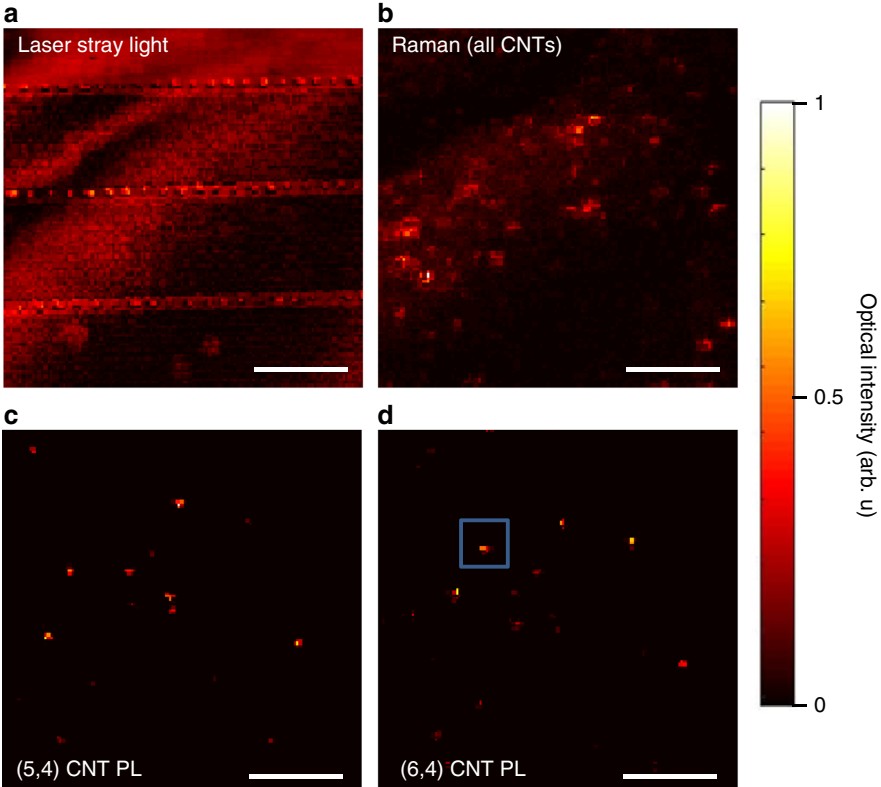

**Fig. 2** Hyperspectral imaging of SWCNT distribution. **a** Laser stray light scanning off of sample surface revealing structural features similar to Fig. 1c and also signatures of deposited material. **b** Hyperspectral Raman imaging of same sample surface area, capturing the G-mode phonon of all SWCNTs. **c** Hyperspectral PL map of same area revealing (5, 4) SWCNTs filtered at $855 \pm 10$ nm. **d** Hyperspectral PL map of same area revealing (6, 4) SWCNTs filtered at $880 \pm 10$ nm. Data are recorded at a sample temperature of 3.8 K. Scale bars are 10 μm

## Results

**Design and characterization of plasmonic nanocavities.** With the goal of combining the superior optical properties of co-polymer wrapped SWCNTs featuring long exciton coherence times[24] and the unmatched light enhancement properties of plasmonic nanocavities featuring gap-modes, we designed dense dimer nanogap antenna arrays where the SWCNT is located on top (Fig. 1a). To guide the structural design of the plasmonic chips we carried out finite-difference time domain (FDTD) simulations to achieve spectral matching of emitter and mode (Supplementary Fig. 1). As shown in Fig. 1b, the intensity enhancement provided by the gap mode can be three orders of magnitude. However, the actual measured EF of the exciton emission can be significantly lower, predominantly due to the dominant NR recombination of excitons in SWCNTs that need to be overcome by the Purcell enhanced radiative rate. Furthermore, EF is reduced due to spatial mismatch between emitter and mode, lateral orientation mismatch of one-dimensional (1D) SWCNTs, variations of the exciton emission energy with nanotube chirality, short circuit of the plasmon mode through residual SWCNT conductivity, and PL quenching due to metal loss.

Several techniques allow us to systematically overcome these challenges, resulting in efficient emitter mode coupling. First of all, the bowtie antenna arrays, fabricated via electron beam lithography (Fig. 1c–e), were covered with a 2 nm thick $Al_2O_3$ spacer layer. This provides the required distance from the gap mode to achieve dominant radiative recombination and minimizes metal loss of the dipole emitter[18]. Another important property of the $Al_2O_3$ spacer is the elimination of charge disorder in the dielectric environment, strongly reducing spectral diffusion (SD) of excitons, as we recently demonstrated for quantum

emitters in boron nitride[25]. In addition, the dense plasmonic array with an antenna spacing of 600 nm matching the average SWCNT length (Fig. 1d) provides a light collection efficiency of 64% that is maintained even for large spatial detuning (See Supplementary Fig. 2). This results in a light collection efficiency enhancement factor of up to $\varepsilon = 2$ (average $\varepsilon = 1.7$), when compared to the collection efficiency of reference SWCNTs located on planar metal mirrors (32%), or $\varepsilon = 5.1$ when compared to bare SWCNTs on glass (13%). By varying the geometry parameters of the nanocavity one can achieve spectral resonance which is rather straightforward since the typical Q-factors of plasmonic modes are rather low ($Q = 2$–20)[26], enabling the simultaneous coupling of exciton absorption and emission dipoles to the cavity mode, as demonstrated experimentally in Fig. 1f. To drastically enhance the chance that individual semiconducting SWCNT with known chirality and emission wavelength are spectrally coupled we have synthesized small-diameter SWCNTs via laser-vaporization (see Methods). Co-polymer (PFO-BPy) wrapping of these SWCNTs[5] produces dispersions with large numbers of (5, 4), and (6, 4) semiconducting SWCNTs with emission wavelengths well-positioned for coupling to the cavity resonance. The SWCNT dispersions fabricated in this way were dried out on the plasmonic chips.

**Spatial map and quantum light emission of individual SWCNTs.** Figure 2a shows a two-dimensional (2D) spatial scan of stray light off of the sample surface revealing the outline of the markers and bowtie arrays, while it shows in addition large regions of enhanced scattered light corresponding to the deposited molecular material. To clearly identify SWCNTs,

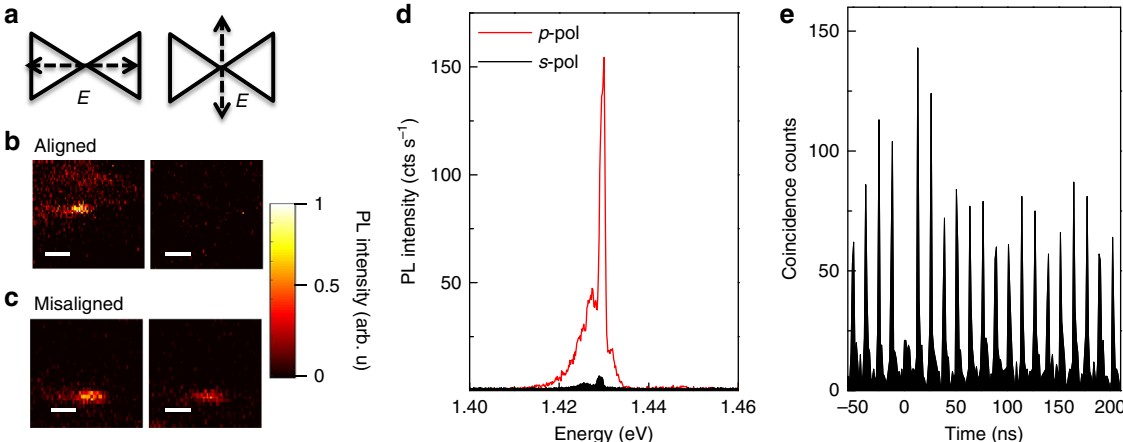

**Fig. 3** Polarization dependence and quantum light signature of exciton emission. **a** Schematic of excitation laser with linear polarization set to be either parallel (p-pol, left) or perpendicular (s-pol, right) to the long axis of bowtie dimer. **b** Photoluminescence (PL) maps of individual SWCNT that are well-aligned along the dimer axis for both p-pol (left) and s-pol (right) excitation. **c** Similarly recorded polarization-dependent PL maps for a SWCNT that is orientationally misaligned. Scale bars are 1 μm. **d** Corresponding PL spectra for a well-aligned SWCNT featuring a large excitation polarization extinction ratio of 25:1. **e** Second-order correlation function $g^{(2)}(\tau)$ recorded at $P_{exc} = 200\,\mu W$ excitation power demonstrating pronounced single photon antibunching with $g^{(2)}(0) = 0.30 \pm 0.06$

hyperspectral images were recorded for the Raman G band signal, which is found regardless of tube chirality. The corresponding 2D scan in Fig. 2b reveals bright spots corresponding to individual SWCNTs at a moderate density in about half of the area. To clearly address the subset of either (5, 4) or (6, 4) SWCNTs we have carried out hyperspectral PL mapping in Fig. 2c, d covering the corresponding exciton PL emission bands located at 855 and 880 nm, respectively. Individual SWCNTs appear as nearly spherical dots as expected, since the SWCNT length (0.5–1 μm) is close to the excitation laser spot size. In this way, individual SWCNTs with known chirality can be addressed for further investigations with respect to their coupling strength to the plasmonic mode (blue box in Fig. 2d).

To create strong Purcell enhancement the lateral orientation of the SWCNT ideally matches both the pump polarization for efficient light absorption, as well as the plasmon mode polarization for efficient cavity coupling. Simulations show that when the incident polarization of the pump laser is parallel to the long axis of the bowtie structure (p-pol), the field enhancement is two orders higher as compared to the orthogonal direction (s-pol), a condition easy to fulfill since the orientation of the bowties is known from imaging (Fig. 1c–e). To probe for the subset of SWCNTs that are oriented along the bowtie dimer axis we recorded hyperspectral maps for s-pol and p-pol excitation (Fig. 3). While some SWCNTs are found that display only a weak intensity variation and are thus either spatially away from the hot-spot or orientationally misaligned, others clearly switch off with s-pol excitation. Several SWCNTs are found displaying a rather pronounced PL polarization extinction contrast with one example of 25:1 following the plasmon mode orientation within a few degrees (Fig. 3d). This strong PL extinction ratio is in contrast to SWCNTs that are located on bare wafers with values varying from 3:1 to 5:1[27]. The significantly stronger polarization contrast demonstrated here is a direct indication of exciton-plasmon mode coupling. In addition, Fig. 3e demonstrates pronounced photon antibunching signatures of the nanocavity coupled exciton emission. In this case, we observe a second-order correlation function of $g^{(2)}(\tau = 0) = 0.30 \pm 0.06$ when recorded under pulsed excitation near saturation ($P_{exc} = 200\,\mu W$) with a 10 nm broad bandpass filter and normalized to the average peak area beyond the first two side peaks that are affected by bunching. Further improvements to the single-photon purity could be achieved by narrowband filtering, for example with a spectrometer, or by coupling to an on-chip waveguide[4] or dielectric cavity mode[7]. The observed values are nevertheless already significantly below $g^{(2)}(\tau = 0) = 0.5$ and thus clearly prove the quantum-dot like (0-dimensional) nature of the exciton emission[28], while the bunching signature is comparable to our findings for SWCNTs in dielectric cavities in the presence of blinking at the ns time scale[27].

**Time-integrated measurements.** While the strong polarization contrast is a good indicator for exciton-plasmon coupling, the quantitative strength can be characterized by the intensity enhancement factor EF. In the saturation regime EF has contributions from three processes, an enhanced absorption rate $\alpha$, enhanced light extraction $\varepsilon$, and the total rate enhancement factor $\gamma_{on}/\gamma_{off}$ from the Purcell effect, which itself includes radiative and NR rates as well as metal loss (Supplementary Eqs. 7–9), resulting in EF = $\alpha\varepsilon\gamma_{on}/\gamma_{off}$. We define coupled SWCNTs as those spectrally matched with the plasmon mode and located on the bowtie arrays, but not necessarily fully laterally or orientationally aligned to the high field area of the mode. In contrast, we define reference SWCNTs as those that are found far from the $100 \times 100\,\mu m^2$ patterned areas on the planar metal with 60 nm $SiO_2$ and 2 nm $Al_2O_3$ spacer. For recording light-light curves from the reference SWCNTs we have in every case aligned the pump laser polarization with the nanotube axis to maximize emission. For the coupled SWCNTs the laser polarization was fixed along the bowtie long axis to efficiently excite the gap-mode that is also polarized along this axis. Figure 4a displays the integrated intensity vs. pump power comparing reference (blue dots) and coupled SWCNTs (red dots) that increases linearly until slight saturation sets in at the highest pump powers. This behavior can be modeled with a three-level rate equation analysis of 0D quantum-dot like excitons (Supplementary Fig. 3).

To determine EF from the data in Fig. 4a we measured the emission intensity from 20 reference SWCNTs away from the nanogap arrays. Figure 4b shows the histogram of recorded intensity at a fixed pump power of 0.6 mW resulting in an average value of 920 cts s$^{-1}$ and a standard deviation of 170 cts s$^{-1}$, corresponding to only 20.7% variation. The intensity variation of the reference SWCNTs is remarkably stable, which is a result of

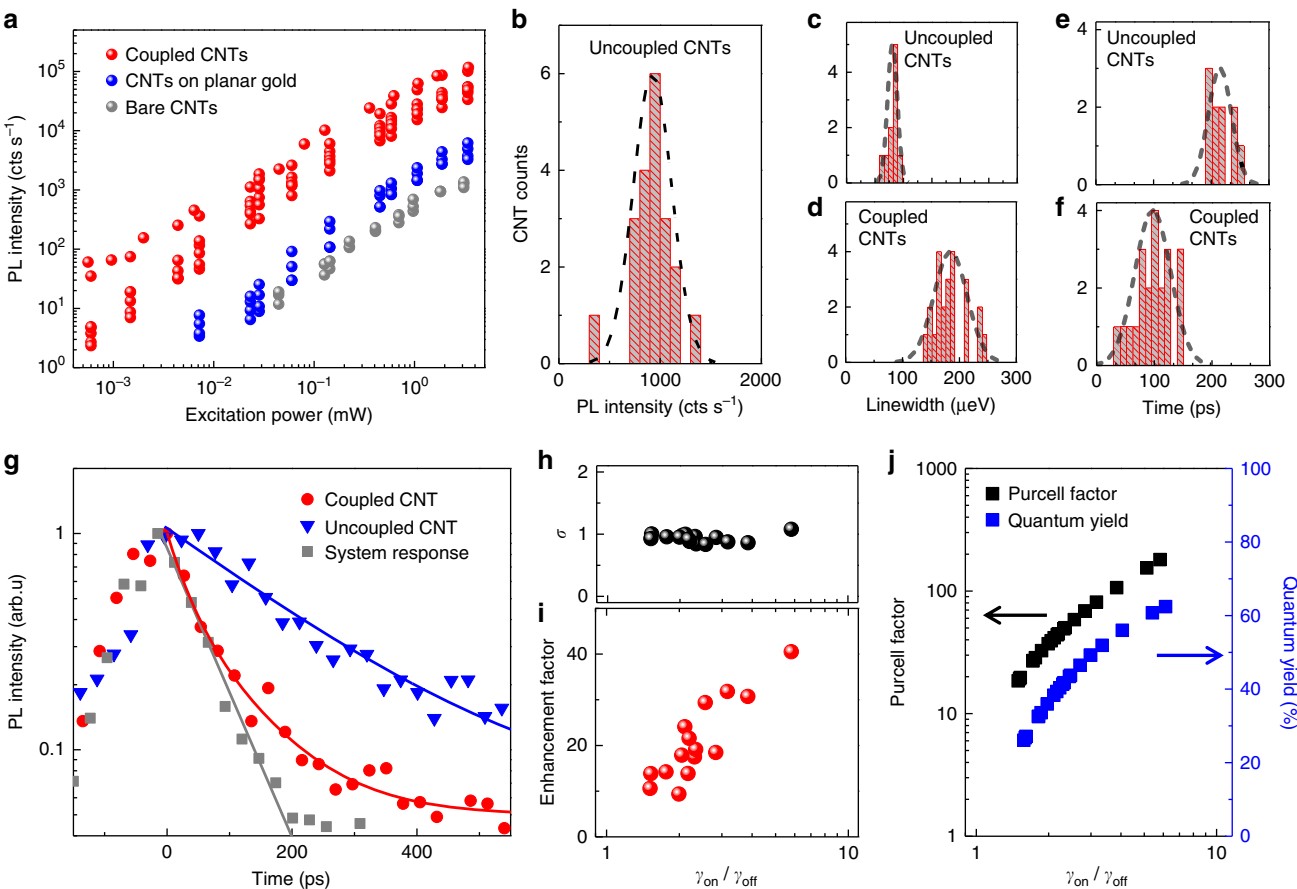

**Fig. 4** Quantifying SE enhancement of the $E_{11}$ exciton emission. **a** Integrated PL intensity of the $E_{11}$ zero-phonon line (ZPL) as a function of excitation power. Red circles are data for coupled SWCNTs, blue circles for reference SWCNTs (off-cavity), and gray circles for bare SWCNTs on Si/SiO$_2$ substrate. **b** Histogram of ZPL peak intensity for 20 reference SWCNTs recorded at fixed excitation power of 0.6 mW. **c**, **d**, Histogram of Lorentzian full width half maximum (FWHM) values of the $E_{11}$ ZPL for 10 reference SWCNTs (**c**) and 21 coupled SWCNTs (**d**). **e**, **f** Histogram of lifetimes for 10 reference SWCNTs (**e**) and 21 coupled SWCNTs (**f**). Raw data are shown in Supplementary Note 6. **g** Temporal dynamics of $E_{11}$ exciton emission recorded by TCSPC at 200 µW excitation power. Gray squares: System response for back-reflected laser light. Solid gray line: Mono-exponential fit representing the system response. Blue triangles are data for a reference SWCNT and the solid blue line is a deconvolved fit which yields a mono-exponential decay time of $\tau_{off} = 248 \pm 3$ ps. The red circles are data from a coupled SWCNT and yield $\tau_{on} = 37 \pm 3$ ps (red solid line). **h** Ratio $\sigma$ representing the total rate enhancement factor ($\gamma_{on}/\gamma_{off}$) determined from the time-integrated approach divided by $\gamma_{on}/\gamma_{off}$ measured directly via time-correlated single photon counting (TCSPC). The strong correlation, $\sigma \sim 1$, indicates that both techniques determine the same physical quantity. **i** Correlation between intensity enhancement factor (EF) and $\gamma_{on}/\gamma_{off}$ from TCSPC, which indicates dominant radiative recombination. **j** Underlying Purcell factor (black square) and quantum yield (blue square) as a function of measured $\gamma_{on}/\gamma_{off}$. All data are recorded at 3.8 K

the PFO-BPy protection of the exciton as well as the Al$_2$O$_3$ passivation layer, resulting in a good reference to determine EF (see Supplementary Note 5). When comparing SWCNTs coupled to the plasmonic gap modes to the average value of the reference SWCNTs we obtain peak values of EF = 40 (average EF = 18 ± 7, lowest EF = 8). Likewise, peak values of EF = 98 (average EF = 44, lowest EF = 20) are found when comparing to bare SWCNTs.

Apparently, coupled SWCNTs display 1–2 orders of magnitude higher emission intensity compared to reference SWCNTs, indicating that virtually all emitters on the plasmonic array are coupled, regardless of orientation/location. This omnipresent enhancement is expected since the SWCNT length matches the nanoantenna spacing and $\varepsilon$ is on average 1.7 (Supplementary Fig. 2). The enhanced light collection from $\varepsilon$ is nevertheless the smallest contribution to EF = $\alpha\varepsilon\gamma_{on}/\gamma_{off}$. A larger contribution originates from $\alpha$ that can be determined for each individual SWCNT from the pump induced linewidth broadening in Fig. 5a (vide infra). Figure 4c shows histograms of 10 reference SWCNTs recorded at fixed excitation power of 25 µW that display a remarkably well-defined average zero-phonon linewidth (ZPL) of

$80 \pm 8.6$ µeV, corresponding to a variation of only 10.8%. In contrast, Fig. 4d shows coupled SWCNTs display significantly broader values that vary from $141 \pm 6$ µeV to $247 \pm 6$ µeV, depending on the actual pump-induced absorption of each probed SWCNT. Using this information, one can estimate the underlying total rate enhancement $\gamma_{on}/\gamma_{off}$ from the time integrated experiment for each coupled SWCNTs. For example, in the best case for EF = 40 and $\varepsilon = 1.7$ and $\alpha = 3.8$, we find the total rate enhancement factor up to $\gamma_{on}/\gamma_{off} = 6.3 \pm 0.8$ (630%), implying that the dominant contribution to EF originates from the underlying Purcell effect. As a result, the measured photon emission rate under continuous-wave excitation from coupled SWCNTs reaches up to 120,000 cts s$^{-1}$, which after correcting for the detection system efficiency of $0.77 \pm 0.2$% (Supplementary Note 4) yields 15.6 MHz emitted into the first lens.

**Time-resolved measurements.** The time-integrated approach of measuring EF is an indirect way to determine the rate enhancement $\gamma_{on}/\gamma_{off}$ and relies on knowledge of light collection efficiency

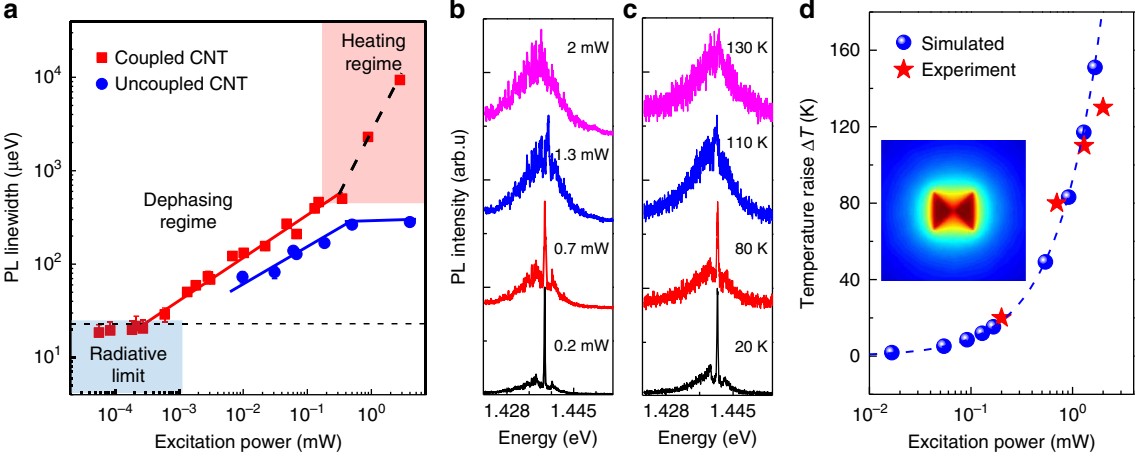

**Fig. 5** Excitation power and temperature dependence of the $E_{11}$ exciton emission linewidth. **a** Spectral linewidth determined from Voigt deconvolution fits (Supplementarty Note 7) as a function of laser excitation power (780 nm) for a plasmonically coupled SWCNT (red squares) and reference SWCNT (blue circles). The blue box highlights the pump regime when the linewidths reaches the Purcell-enhanced radiative lifetime limit of the emitter ($T_1 = 37$ ps $= 18$ μeV). Gray dashed line illustrates spectral resolution limit (23 μeV). The red box highlights the pump regime where plasmonic heating affects the emission spectrum. Data are recorded at 3.8 K. **b** PL spectra for the coupled SWCNT recorded in the plasmonic heating regime where a significant lattice temperature increase causes the break-up of the acoutic-phonon confinement leading to strong linewidth broadening. **c** Comparision spectra for the coupled SWCNT recorded under moderate pump powers of 200 μW and for increasing temperature from 20 to 130 K. **d** FDTD simulation of temperature raise $\Delta T$ of the bowtie structure as a function of excitation power (blue dots). Inset: Simulated heat map of the bowtie antenna. The red stars are the data points from the experiments in **b** with the temperature equivalent taken from the experiment in **c**

and enhanced absorption. To directly assess the rate enhancement of the coupled SWCNTs we have determined $\gamma_{on}/\gamma_{off}$ from time-resolved analysis utilizing time-correlated single photon counting (TCSPC). Figure 4g shows two exemplary decay traces for a reference and a coupled SWCNT that have been deconvolved using the system response function. The extracted mono-exponential decay times are $\tau_{off} = 248 \pm 3$ ps and $\tau_{on} = 37 \pm 3$ ps, respectively. The histogram in Fig. 4e demonstrates that the reference SWCNTs display decay times that vary in a rather narrow range from 194 to 248 ps with an average value of $\tau_o = 215$ ps and a standard deviation of 20 ps, which corresponds to a variation of only 9.3%. In contrast, every coupled SWCNT displays a significantly faster rate that varies from $155 \pm 3$ ps to $37 \pm 3$ ps in the best case (Fig. 4f). By normalizing to the average value of the reference SWCNTs (215 ps), which represents $\gamma_{off}$, one can determine $\gamma_{on}/\gamma_{off}$ for each of the coupled SWCNTs. For the best case the TCSPC experiment yields a 6-fold total rate enhancement $\gamma_{on}/\gamma_{off} = 5.8 \pm 0.5$ (580%) that matches within error the value of $6.3 \pm 0.8$ (630%) determined from the time-integrated approach for this particular SWCNT. Figure 4h plots the correlation parameter $\sigma$, which represents the ratio of rates from the time-integrated and time-resolved experiments for all investigated SWCNTs. The close proximity to unity, $\sigma = 0.94 \pm 0.07$, shows excellent consistency of these two methods to determine $\gamma_{on}/\gamma_{off}$.

In addition, we find a clear correlation between rate increase and intensity enhancement as demonstrated in Fig. 4i, showing that once the rate gets significantly affected by the Purcell enhancement it is predominantly of radiative nature. A possible increase in NR recombination due to an underlying dominant metal loss in plasmonics would create a decrease of EF with increasing $\gamma_{on}/\gamma_{off}$, which is not observed in our case. This finding is also consistent with recent reports on emitters with unity quantum efficiency (negligible NR vs. radiative rate) placed in close proximity (2 nm vertical spacer) to nanocavity gap modes, showing that emission is predominantly radiative (75%), while 25% is attributed to the metal loss $\gamma_M$, which is a property of the plasmonic nanocavity that is independent of the emitters[18].

Such a high (6-fold) total rate enhancement dominated by radiative recombination is quite remarkable, particularly in light of recent reports of PFO-wrapped SWCNTs coupled to high-Q dielectric cavities in the Purcell regime that show measured total rate enhancement of only 10%[7]. It is important to note that SWCNTs suffer in general from rather low quantum yield $\eta_{off} = \frac{\gamma_R}{\gamma_R + \gamma_{NR}}$ of 2–7%[5–9], where $\gamma_R$ and $\gamma_{NR}$ are the radiative and non-radiative decay times of the reference system. Several recent studies arrive at PL quantum yield estimates of ~2% for the case of PFO-wrapped SWCNTs that have undergone sonication[7–9], a 49-fold faster NR rate compared to the radiative rate in the absence of a Purcell effect. Using a value of $\eta_{off} = 2 \pm 0.5\%$, determined from single nanotube studies via single-photon emission saturation, Jeantet et al.[7] determine an underlying $F_P = 5 \pm 2$ for the measured 10% total rate enhancement. In this case most of the Purcell enhancement of $\gamma_R$ is used to catch up with $\gamma_{NR}$. Even at one order of magnitude enhancement of $\gamma_R$ the total measured rate $\gamma_{on}$ increases only by a few percent, while $F_P = 50$ would just double the measured rate $\gamma_{on}$. Assuming our PFO-wrapped SWCNTs also have $\eta_{off} = 2\%$ for the reference case, and suffer an additional 25% metal loss when they are coupled, one can calculate the underlying Purcell factor from the relation: $F_P = 0.75 \left( \frac{\gamma_{on}}{\gamma_{off}} - 1 \right) \eta_{off}^{-1}$ (Supplementary Eqs. 7–15). As shown in Fig. 4j, the measured $\gamma_{on}/\gamma_{off}$ rates correspond to an underlying $F_P = 180$ for the best case, an average $F_P = 57$, and a minimum of $F_P = 19$. This is the highest Purcell factor reported for SWCNTs, that is threefold larger compared to a very recent report by Jeantet et al. ($F_P = 60$) for an optimized dielectric scanning-fiber cavity[29]. Nevertheless, the exciton emission from SWCNTs still underperforms the theoretical value for our plasmonic nanocavities ($F_P = 6591$), predominantly due to orientational and/or spatial mismatch, leaving room for future improvements using deterministic assembly. Such a high $F_P$ implies that the SE coupling factor $\beta = F_P/(1 + F_P)$ is unity (99.5%), indicating that virtually all of the SE emission is coupled to the nanocavity mode[30].

The Purcell enhanced emitter has furthermore a cavity-enhanced quantum yield $\eta_{on}$ that can be significantly larger compared to the reference system, since the Purcell enhanced radiative rate catches up with all NR losses in the system. Ultimately $\eta_{on}$ cannot reach exactly unity in plasmonic systems since the metal loss cannot be entirely avoided. At best, for high enough Purcell factors ($\gamma_R \gg \gamma_{NR}$), $\eta_{on}$ saturates at values determined by the metal loss rate $\gamma_M$. The estimated cavity-enhanced quantum yields for each SWCNT are shown in Fig. 4j (Supplementary Equation 16). The average quantum yield of the 21 coupled SWCNTs is 42%, including the worst case of 26% and the best case of $\eta_{on}$ = 62%. As a result, every SWCNT on the plasmonic chip is enhanced by at least one order of magnitude in intensity, Purcell factor and quantum yield, while $\eta_{on}$ is up to 30-fold improved compared to the reference SWCNTs approaching near-unity.

**Ultra-narrow linewidth regime.** The demonstrated large light enhancement and improved internal quantum efficiency enables one to study the exciton photophysics in two extreme regimes—at ultra-low optical pump powers where pump-induced exciton dephasing is minimized as well as at high pump powers where plasmonic heating is expected. Specifically, we have studied the ZPL of the $E_{11}$ exciton emission over five orders of magnitude variation in excitation pump power (Fig. 5a). For the reference SWCNT a ZPL of 75 µeV is found at 10 µW pump power. This is almost twice better than the previous record for PFO-SWCNT material[24] and is a result of the 2 nm thick $Al_2O_3$ layer that strongly reduces spectral diffusion typically found when SWCNTs touch $SiO_2$ or glass substrates[27]. At lowest pump powers of 50 nW the plasmonically coupled SWCNT displays an ultra-narrow ZPL of only $18 \pm 3$ µeV shown in Fig. 5a. Note that all measured spectral linewidths of the exciton emission are broader than the measured spectrometer resolution for laser light (23 µeV) and values that characterize the additional contribution from the ZPL have been extracted by a standard Voigt-deconvolution (see Supplementary Note 7). This linewidth is the narrowest ever reported for carbon nanotube excitons, 4 to 7-fold narrower than previous best reports for PFO-wrapped SWCNTs[24,31] and twice narrower than the resolution-limited linewidth reported for surfactant-free air-suspended SWCNTs[32]. Importantly, the measured Purcell enhanced radiative lifetime value from the time-resolved experiment suggests that a significant part of the emitter linewidth is provided via lifetime broadening. The measured rate of 37 ps corresponds to $\Delta E = 17.8$ µeV, which matches within the error with the value determined independently from the time-integrated linewidth analysis. The effect that the linewidth levels off in Fig. 5a at lowest pump powers (blue shaded area) is thus a direct manifestation of a Purcell-enhanced emitter reaching its radiative linewidth limit. Using the well-known relation $FWHM = 2\Gamma = \frac{2\hbar}{T_2} = \frac{\hbar}{T_1} + \frac{2\hbar}{T_2^*}$, where FWHM is the full width at half maximum of the ZPL shown in Fig. 5a, one can further estimate the $T_2$ coherence time of the exciton emission to be at least 73 ps as the lower bound.

**Plasmonic heating regime and single-molecule thermometry.** At the other extreme of high pump power, we demonstrate that plasmonically induced heating can be measured in the near-field based on the peculiar changes in the exciton spectrum of an individual PFO-wrapped SWCNT. With increasing pump power the acoustic-phonon wings that peak about 2 meV symmetrically around the ZPL strongly gain in intensity until they fully merge with the ZPL (Fig. 5b). In total, the pump induced broadening and plasmonic heating contribute to a 560-fold spectral broadening of the exciton emission from $\Gamma = 18$ µeV to $\Gamma = 10$ meV,

when the pump power is raised from 50 nW to 2 mW. In contrast, the reference SWCNT is unaffected by the plasmonic heating and saturates at about $\Gamma = 200$ µeV at highest pump powers (Fig. 5a). Similar spectral evolution can be observed by exciting the coupled SWCNT only moderately (200 µW), but with progressive increase of the base temperature, as shown in Fig. 5c. It is thus clear that the spectral changes accompanying high power (>300 µW) excitation of the coupled SWCNT are associated with plasmonically induced heating. Our detailed exciton lineshape analysis in ref. 24 (involving the temperature dependent boson occupation factor, see Supplementary Note 8) demonstrated that this spectral evolution is caused by the thermal break-up of the acoustic-phonon confinement provided by the co-polymer backbone of the PFO-wrapped SWCNTs. This model allows us to extract the plasmonically induced rise in temperature at the single-nanotube level. The extracted single-molecule read-out matches well with thermal transport modeling of the total heat power confined on the bowtie antenna, reaching 150 K at 2 mW optical pump power (Fig. 5d). Perfect spatial matching for thermal coupling is apparently easier to achieve as compared to the case of Purcell enhancement, which is also evident from the heat map in Fig. 5d showing that the entire area of the bowtie antenna (250 nm side length) contributes to heating.

## Discussion

We note that unlike previous demonstrations of measuring heat in plasmonic nanostructures[33], we directly determine here the thermal near-field through an individual molecule rather than a distributed ensemble in a thin film. The utility of PFO-wrapped SWCNTs as single-molecule thermometers with all-optical read out is nevertheless limited to the temperature range of 4–120 K, above which the acoustic phonon barriers thermalize and the exciton linewidth is rendered broadband. The readout sensitivity of the temperature is about $\pm 5$ K. The sensitivity is limited by the required lineshape analysis to extract temperature from the Bose–Einstein occupation factor, which determines the intensity ratio of the ZPL to the phonon side-peak intensity[24]. Nevertheless, these experiments uncover the unique interplay of excitons, phonons, and plasmons at the nanoscale.

Our results further reveal a new way to characterize the enhanced absorption rate in plasmonically coupled systems. Figure 5a shows that at any given pump power $P$ the coupled SWCNT has a larger ZPL in comparison to the reference SWCNT by a factor 1.75–1.96. The additional ZPL broadening can be attributed to an effectively higher excitation power due to the enhanced absorption provided by the plasmonic mode. The magnitude can be quantified since the coupled SWCNT in Fig. 5a displays a clear trend over three orders of magnitude in $P$ that fits to a square root function, $\Gamma \propto \sqrt{P}$. Using this functional dependence, one can estimate a cavity enhanced absorption from the linewidth data to result in $\alpha = 3.8 \pm 0.2$ for this case. The microscopic mechanism for pump-induced broadening in 1D SWCNTs is often attributed to exciton-exciton scattering (EES)[34]. Our observation of strong antibunching precludes EES as the dominant contribution to the ZPL since the number of excitons is well below 2 ($g^2(\tau = 0) \leq 0.5$) even at $P = 200$ µW (Fig. 3e). It is further apparent that within the dephasing regime (Fig. 5a) the ZPL shape remains Lorentzian, implying that the dominant contribution is from a homogenous dephasing process such as random telegraph noise. Interestingly, the observed $\sqrt{P}$ dependence of $\Gamma$ is similar to the behavior of 0D excitons in quantum dots that are known to be governed by fluctuating charges in the exciton vicinity that also follows a telegraph-noise leaving the linewidth Lorentzian[35]. Although the exact microscopic mechanism remains elusive in PFO-wrapped SWCNTs, our

analysis shows that the pump power-induced dephasing is a useful resource to quantitatively determine enhanced absorption rates in cavity-coupled systems.

As opposed to molecular dyes with ubiquitous photobleaching, the SWCNTs emitters demonstrated here are characterized by temporally stable emission and narrow distribution in terms of their optical properties and are thus particularly viable for the practical realization of hybrid molecular-nanoplasmonic systems that can boost the photon extraction efficiency for emitters with initially low quantum yields. The advances demonstrated in our study could furthermore enable quantum information protocols on-chip that go beyond single photons, such as entanglement protocols that require indistinguishable photons. To this end the, the ultra-narrow linewidth combined with the large SE Purcell enhancement provided by the nanocavity is promising towards future generation of indistinguishable photons from SWCNTs. However, to become practical, the emitted rate of single photons with optical properties near the lifetime limit needs to be significantly improved, which could be achieved by protecting excitons against pump-induced dephasing when trapped in the energetically deep localization potentials of solitary dopant atoms[3,36]. While perhaps less obvious, enhancing radiative efficiency also has a direct impact on improving the power conversion efficiency of solar energy harvesting devices (including photovoltaics and solar fuels devices), due to the generalized optoelectronic reciprocity theorem[37]. In particular, non-radiative decay reduces the maximum attainable open-circuit voltage for a photovoltaic solar cell. Thus, any solar energy harvesting application benefits directly from approaches that provide either better fundamental understanding of, or technological strategies to improve, the ultimate radiative efficiency of a given semiconductor.

## Methods

**Plasmonic chip fabrication**. The bowtie arrays were fabricated at the CUNY-ASRC facility by electron-beam lithography (EBL) using 1:3 diluted Methyl Styrene/Chloromethyl Acrylate Copolymer (ZEP520A) in anisole that was spin-coated at 2500 rpm onto the Si/SiO$_2$ substrate. The side lengths of the individual bowtie antennas triangles were 250 nm, with a height of 30 nm and gap size varying from 10 to 30 nm. The samples were subsequently patterned in an Elionix ELS-G100 EBL system and developed at a chilled temperature of 2 °C in n-Amyl acetate for 120 s. To convert the polymer template into a plasmonic array we deposited a 3 nm Cr adhesion layer and 30 nm Au metal on the samples in an electron beam evaporator (AJA Orion 3-TH) followed by liftoff in Dimethylacetamide solvent at room temperature. Finally, a 2 nm thick layer of Al$_2$O$_3$ was deposited by atomic layer deposition.

**Carbon nanotube synthesis and dispersion**. SWCNTs with the natural isotope ratio (99%$^{12}$C) were synthesized by the laser vaporization (LV) process, as described previously[38]. The small-diameter SWCNTs synthesized for this study were produced at a furnace temperature of 800 °C in the LV process, and all syntheses were run at a power density of ~100 W cm$^{-2}$ ($\lambda$ = 1064 nm, Nd:YAG). The target consisted of Alfa Aesar graphite (2–15 μm, stock #14736, and 3 wt% each nickel and cobalt catalysts. SWCNTs grown in this way were dispersed in poly [(9,9-dioctylfluorenyl-2,7-diyl)-alt-co-(6,60-{2,20-bipyridine})] (PFO-BPy). Briefly, 1 mg mL$^{-1}$ of raw LV soot was mixed into a solution of 2 mg mL$^{-1}$ of PFO-BPy in toluene. This solution was then sonicated with a 1/2 in. probe tip for 30 min at 40% power (Cole-Parmer CPX 750) in a bath of cool (18 °C) flowing water for heat dissipation. The indicated power output was 28 W, yielding a power density of about 22 W cm$^{-2}$ delivered to the dispersion. After sonication, solutions were centrifuged at 30,000×$g$ for 5 min using a SW32-Ti rotor (Beckman). Finally, PFO-BPy dispersed LV SWCNTs in toluene were deposited directly onto the array region on the sample followed by 105 °C baking on hot plate for 3 h.

**Photoluminescence spectroscopy**. Micro-photoluminescence (μ-PL) measurements were taken inside a closed-cycle cryogen-free cryostat with a 3.8 K base temperature and ultralow vibration (attodry1100 by attocube). Samples were excited with a laser diode operating at 780 nm in continuous wave mode. A laser spot size of about 0.85 μm was achieved using a cryogenic microscope objective with numerical aperture of 0.82. The relative position between sample and laser spot was adjusted with cryogenic piezo-electric $xyz$-stepper while 2D scan images were recorded with a cryogenic 2D-piezo scanner (attocube). Spectral emission

from the sample was collected in a multimode fiber, dispersed using a 0.75 m focal length spectrometer, and imaged by a liquid nitrogen cooled silicon CCD camera. For high resolution measurements we used an 1800-groove grating and 10 microns slit width settings. Laser stray light was rejected using an 800 nm long-pass edge filter. For time-resolved PL lifetime measurement light from a supercontinuum laser (NKT photonics) operating at 78 MHz repetition rate and 7-ps pulse width was filtered by a 10 nm bandpass filter centered at 780 nm. TCSPC experiment was carried out with a coincidence counter (SensL) and a fast avalanche photodiode (APD) with a timing jitter of 39 ps (IDQuantique). The system response function was measured sending scattered laser light from the sample surface with the 800 nm filter removed and at an APD count rate that matches the rate of the $E_{11}$ exciton emission. The second-order correlation function $g^{(2)}(\tau)$ was recorded with the same TCSPC setup but in Hanbury–Brown and Twiss configuration where the exciton emission is first sent through a 10 nm bandpass filter followed by a 50:50 multimode fiber splitter before reaching the two APDs.

**Plasmonic resonance measurement**. To determine the transmission/scattering spectrum of the plasmonic bowtie arrays we have fabricated samples on glass substrate with equivalent dimensions to the ones on Si/SiO$_2$ substrates used for the SWCNT studies but at rather low density of bowties being spaced out at least 5 μm to avoid plasmonic lattice effects[26]. Transmission measurements were carried out under perpendicular incidence excitation with a tungsten-halogen white light source filtered by a broadband linear polarizer. Passing the polarizer the white light was focused onto the samples by a 100× microscope objective and the transmitted light through the sample was collected by a 50× microscope objective and sent through a variable spatial filter before coupling into a multi-mode fiber attached to a high-resolution grating spectrometer with TE-cooled CCD camera.

**Data availability**. The data that support the findings of this study are available from the corresponding author upon request.

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

## Acknowledgements

We like to thank Milan Begliarbekov for supporting the EBL process development at the City University of New York Advanced Science Research Center (ASRC) nanofabrication facility. S.S. and J.H. acknowledge financial support by the National Science Foundation (NSF) under award DMR-1506711 and DMR-1507423, respectively. S.S. acknowledges financial support under NSF award ECCS-MRI-1531237. J.B. and K.M. gratefully acknowledge funding from the Solar Photochemistry Program of the U.S. Department of Energy, Office of Science, Basic Energy Sciences, Division of Chemical Sciences, Geosciences and Biosciences, under Contract No. DE-AC36-08GO28308 to NREL

## Author contributions

S.S. and Y.L. designed the experiment and analyzed the data. Y.L., K.S and C.Z. performed the optical experiments. Y.L. and E.D.A fabricated the plasmonic chips and together with Y.M. contributed theory analysis. K.S.M. and J.L.B. have grown and dispersed SWCNT material. S.S., J.H., J.L.B. and Y.L. co-wrote the paper. All authors discussed results and commented on the manuscript.

## Additional information

**Competing interests:** The authors declare no competing financial interests.

