## [Peer Review File · Nature Communications]

EDITORIAL COMMENT: This paper had been previously reviewed at another journal not participating in a Transparent Peer Review scheme. Please find below the reviewer comments & author responses for this paper while at Nature Communications.

Reviewers' comments:

Reviewer #1 (Remarks to the Author):

Y. Luo et al. show original results on Purcell brightening of carbon nanotubes by means of gap modes of plasmonic resonators. There are two main -remarkable- achievements in this work : one on the material side, the other of the nanotechnology of gapped plasmonic resonators.

The first achievement is an amazingly homogeneous distribution of carbon nanotubes in terms of optical properties (lifetimes, linewidths, brightness...). This sample also brings record linewidths due to strong suppression of spectral diffusion. In my opinion, the authors should put more emphasis on this original aspect of their work. These record linewidths are indeed close to the radiative limit.

(Nevertheless, in their comparison with the state of the art, the authors should mention the work by J. Alexander-Webber et al. Nanoletters 14, 5194 (2014) where comparable values were obtained).

The second aspect of the work is the efficient coupling of carbon nanotubes to plasmonic resonators, which results in a strong Purcell brightening. The study is based on an extensive statistical survey, which makes the conclusions robust to possible statistical fluctuations from tube to tube.

Nevertheless, the values of F_p and effective radiative yield are all based on a quantum yield value taken from another study made on another type of sample. In view of the sensitivity of quantum yields of PFO wrapped nanotubes on the preparation method (Graf et al. Carbon 105, 593 (2016, fig. 4)), this weakens slightly the quantitative analysis. In addition, the authors should be more cautious in claiming record values. In fact, their discussion and comparison with dielectric cavities should be updated to the recently published data by Jeantet et al. (Nanoletters 7, 4184 (2017)) that show Purcell factors and effective quantum yields of comparable magnitude.

A few technical points have to be corrected before publication :

-page 6 : "This shows that once the rate gets significantly affected by the Purcell enhancement, it is predominantly of radiative nature". This statement is misleading. In fact, as the authors show below, even in the strong brightening regime, the effective quantum yield of the nanotubes is 42% "only", which means that the emitter is still predominantly nonradiative. It is actually a key point that "bad" emitters optimally benefit from Purcell enhancement, whereas "good" emitters are brightened only in the saturation regime.

- a weakness of the approach in estimating the Purcell factor lies in the 0.75 ratio for radiative rate vs metal losses that comes from theoretical simulations. Could the authors try to correlate time-resolved data with count-rate data to make their estimate more solid (i.e. based on experimental data rather than numerical simulations) ?

- page 8 : in the discussion on indistinguishability, the estimate of T_1 is taken from lifetime measurements corrected by the quantum yield, as if T_1 should be the radiative lifetime. I am not sure that this is correct. In the relationship between T_1 , T_2 and T_2^* , T_1 should represent the total population decay, both radiative and nonradiative. The same holds for the HOM visibility. (BTW this makes the predicted visibility even better !)

Finally, a more general comment : since plasmonic resonators are intrinsically broad (in terms of spectral width), why did the author only focus cryogenic measurements ? Their brightening should not degrade significantly at room temperature (in contrast to narrow band dielectric cavities), which would be a very nice achievement.

In conclusion, I think that this work is acceptable for Nature Communications if the previous comments are properly addressed.

Reviewer #2 (Remarks to the Author):

The manuscript by Luo et al reports the modification of carbon nanotube photon emission properties by means of coupling to plasmonic bowtie nanocavities. Specifically, the authors find large Purcell enhancements and increased quantum yields as main effects of the plasmonic cavity on the nanotube emission characteristics. Moreover, they speculate that cavity-coupled nanotubes are sources of single indistinguishable photons. Finally, the authors advocate the potential of hybrid nanotube-cavity systems for all-optical thermometry.

Having read the manuscript, the supplementary information and the communication within the first two rounds of reviews, I'm hesitant to recommend the paper for publication in Nature Communications in its present form. While the manuscript seems to have improved considerably from its initial version to the revised version under present review, I have to raise additional concerns beyond the critical aspects that have been settled in response to previous reviews.

My main concern, in line with referee #3, is related to the discussion of narrow spectral linewidths and photon indistinguishability. First of all, I didn't find any statistics on the distribution of linewidths for the coupled nanotube-cavity systems at lowest excitation powers. Second, I'm not convinced that the authors can extract the linewidth below the resolution limit of their spectrometer ($\sim 25 \mu\text{eV}$). The section in the supplementary information dedicated to the procedure of zero-phonon-linewidth (ZPL) deconvolution states that sub-resolution Lorentzian contributions to the Voigt profile can be determined provided "good signal to noise ratio". This is clearly not the case in the supplementary Fig. 6. Moreover, the deconvolution is probably strongly compromised by the overall non-Lorentzian lineshape of the nanotube photoluminescence as discussed in section 8. of the supplementary information. In my opinion, the fact that the linewidth of the nanotube in Fig. 5a levels off at low excitation powers is nothing else but a consequence of the spectral resolution limit. This by itself is not troubling if the linewidth had not been used to calculate the two-photon interference visibility shown as inset to Fig. 5a (the set of data in the inset is a compromised projection but is presented in a way as if it has been actually measured). My point of criticism boils down to an unjustified over-interpretation of the experimental data. If the photon indistinguishability is a major result of the paper, the authors should prove it by performing Hong-Ou-Mandel measurements.

My second main concern is related to the enhancement of the nanotube absorption. The procedure used by the authors to determine the absorption enhancement is still unclear (see the second report of reviewer #1). On the one hand, the rate equation analysis of a three-level model system as discussed in section 3. of the supplementary information is clearly not applicable as it neglects non-radiative decay. The evolution of the linewidth with the excitation power, on the other hand, is not generic as the ZPL broadening can vary from tube to tube due to different coupling constants to acoustic and optical phonons and different environments of fluctuating charges. Moreover, the connection between motional narrowing effects in semiconductor quantum dots and the power dependence of the ZPL broadening in the present system is highly speculative. Given the uncertainty in the absorption enhancement, the enhancement factor can vary significantly. The authors state that on this basis "one can qualitatively estimate the intensity enhancement" yet they proceed by discussing quantitative numbers.

Finally, the practicability of the coupled nanotube-cavity system for thermometry has not been evaluated at all. What determines the sensitivity of the system, what is the temperature range for useful implementation, how is the readout compromised by simultaneous heating of the plasmonic cavity, etc?

The manuscript, void of the burden of over-interpretation, reports strong Purcell enhancement of a

single luminescent (macro)molecule by a plasmonic bowtie nanoantenna. As opposed to molecular dyes with ubiquitous photobleaching, nanotubes with temporally stable emission are particularly viable for the practical realization of such hybrid molecular-nanoplasmonic systems which boost the photon extraction efficiency for emitters with low quantum yields. I find this setting scientifically sound and the successful realization of such a technologically challenging hybrid system remarkable. Placed into an appropriate context, the extensive results of the comparative study of coupled vs. uncoupled nanotubes would justify, in my opinion, publication in Nature Communication without the need of unjustified projections or overstressed potential for applications.

Reviewer #1

General comment 1: Y. Luo et al. show original results on Purcell brightening of carbon nanotubes by means of gap modes of plasmonic resonators. **There are two main -remarkable- achievements in this work:** one on the material side, the other of the nanotechnology of gapped plasmonic resonators. The first achievement is an **amazingly homogeneous distribution** of carbon nanotubes in terms of optical properties (lifetimes, linewidths, brightness...). This sample also brings **record linewidths** due to strong suppression of spectral diffusion. In my opinion, the authors should put more emphasis on this original aspect of their work. These record linewidths are indeed close to the radiative limit. (Nevertheless, in their comparison with the state of the art, the authors should mention the work by J. Alexander-Webber et al. Nanoletters 14, 5194 (2014) where comparable values were obtained).

Our response: We thank the reviewer for the positive feedback and statements that our work constitutes a remarkable achievement. We agree with the suggestions and have added a more systematic comparison between our work and previously reported works in the revised manuscript, e.g. by including the work by J. Alexander-Webber et al. Nanoletters 14, 5194 (2014), which demonstrated 78 μeV linewidth that is only 4-fold broader compared to our work and only two-fold broader compared to the 40 μeV of the Munich group, and thus largely in line with ultra-narrow linewidth regimes in SWCNTs.

General comment 2: The second aspect of the work is the efficient coupling of carbon nanotubes to plasmonic resonators, which results in a strong Purcell brightening. The study is based on an extensive statistical survey, which makes the conclusions robust to possible statistical fluctuations from tube to tube. Nevertheless, the values of F_p and effective radiative yield are all based on a quantum yield value taken from another study made on another type of sample. In view of the sensitivity of quantum yields of PFO wrapped nanotubes on the preparation method (Graf et al. Carbon 105, 593 (2016, fig. 4)), this weakens slightly the quantitative analysis.

Our response: We thank the reviewer for the statement that the updated extensive statistical data set we had provided in the previous revision is now considered to be robust against fluctuations and thus strongly supports our original claims from first submission to Nature Photonics. Regarding estimates of Purcell factor and quantum yield improvement and the issue of sample fabrication technique, it is important to note that multiple recent studies indicate that a PL quantum yield in the range of 2% is characteristic for SWCNTs that have been extracted with polyfluorene derivatives *via* bath sonication and/or light tip sonication. As discussed in the original manuscript, in ref. 7 and in a related PhD thesis Jeantet et al. showed that the SWCNT QY for PFO-BPy dispersed tubes was 2%, although they do not specify the conditions used to disperse their SWCNTs (i.e. bath or tip sonication, duration, power density, sonicator brand/model, etc.). We have now also added the suggested reference by Graf et al. (Carbon 105, 593 (2016)) which strengthens this point since they show also a QY of $1.8 \pm 0.2\%$ for PFO-BPy wrapped (6,5) SWCNTs using bath sonication. Importantly, they show that tip sonication for a full 5.5 hours only reduces the quantum yield to 1.3%, although they do not indicate the power delivered by their sonicator. We have also added another pertinent reference, Mouri et al. (J. Phys. Chem. C **116**, 10282 (2012)), which allows for the best quantitative comparison with our own study. In this study, they first employ bath sonication and then systematically vary the time over which samples are exposed to tip sonication (from 0 minutes to 800 minutes). They also denote the power density (500 W/cm^2) imparted by their sonicator. In our study, we sonicate our SWCNTs using a $\frac{1}{2}$ -inch tip ($A = 1.27 \text{ cm}^2$) for 30 minutes and measure a power of 28 W

(translating to a delivered power density of 22 W/cm²) In Mouri et al., the PL quantum yield for PFO-wrapped (8,6) SWCNTs was measured to be 2.2% for bath sonicated SWCNTs. When subjecting the same SWCNTs to tip sonication at a delivered power density of 500 W/cm² for 30 minutes, the PL quantum yield dropped to 1.6%. Thus, the duration of tip sonication in this experiment is identical to our own, but the power density is significantly higher (~23x). It is therefore reasonable to expect that the PL quantum yield for our SWCNTs should fall below that of the bath sonicated SWCNTs (2.2%) and above that of the 30 minute tip-sonicated SWCNTs in their experiment (1.6%), but would likely fall above the mean of the two (1.8%) since our sonication power density is significantly lower than that employed in Mouri et al. The close agreement between these three studies for samples that are prepared under the conditions employed in our study supports the reference value of 2% used in the current study.

General comment 3: In addition, the authors should be more cautious in claiming record values. In fact, their discussion and comparison with dielectric cavities should be updated to the recently published data by Jeantet et al. (Nanoletters 7, 4184 (2017)) that show Purcell factors and effective quantum yields of comparable magnitude.

Our response: We realize that between our initial submission of Oct.2016 and current revision Sep.2017 Voisin's group has made significant progress by fabricating dielectric cavities with smaller mode volumes leading to larger Purcell numbers in their very recent work in Nano Letters. We have now included this work and rephrased our claim in the revised manuscript. Purcell factors in our work are nevertheless still three-fold larger than any previous report, i.e. still a record, but more importantly have been achieved for an a priori scalable on-chip system featuring broadband light enhancement, which is quite different to the bulky narrowband cavities in other work.

Technical comment 1: page 6 : “This shows that once the rate gets significantly affected by the Purcell enhancement, it is predominantly of radiative nature”. This statement is misleading. In fact, as the authors show below, even in the strong brightening regime, the effective quantum yield of the nanotubes is 42% “only”, which means that the emitter is still predominantly nonradiative. It is actually a key point that “bad” emitters optimally benefit from Purcell enhancement, whereas “good” emitters are brightened only in the saturation regime.

Our response: Indeed, the word “predominantly” is mathematically incorrect to describe our average value of 42% and only applies for some of the best results with values up to 62%. We have carefully revised the language in this sentence.

Technical comment 2: a weakness of the approach in estimating the Purcell factor lies in the 0.75 ratio for radiative rate vs metal losses that comes from theoretical simulations. Could the authors try to correlate time-resolved data with count-rate data to make their estimate more solid (i.e. based on experimental data rather than numerical simulations)?

Our response: We had indeed taken the number from metal loss from literature of a very similar plasmonic nanogap cavity made by another group. In the revised manuscript, we provide now a detailed FDTD simulation of our exact structure and fabrication parameters with the new Supplementary Fig. 2c. This figure shows the amount of power emitted radiatively from the plasmon mode as a function of emitting wavelength. The added section further supports our Purcell factor estimates and is also in line with the literature findings on these gap-mode systems that we had previously used (75%). Note that quantitative spectroscopy from time-integrated count rate experiments is rather difficult to calibrate towards absolute numbers and carries a large error. We have thus based the Purcell factor estimation on time-resolved data determining relative rates (coupled vs. uncoupled) rather than absolute intensities, which is the most trusted approach to determine rate enhancements in cavity-QED. See for example our work on Purcell effect of QDs in Nature Photonics 1, 704 (2007).

Technical comment 3: page 8: in the discussion on indistinguishability, the estimate of T_1 is taken from lifetime measurements corrected by the quantum yield, as if T_1 should be the radiative lifetime. I am not sure that this is correct. In the relationship between T_1 , T_2 and T_2^* , T_1 should represent the total population decay, both radiative and nonradiative. The same holds for the HOM visibility. (BTW this makes the predicted visibility even better!)

Our response: Note that we decided to remove this section on HOM visibility values from the manuscript in light of the comments given by reviewer #2, who correctly pointed out that our HOM visibility figure is just a theoretical expectation value but not directly proven in an experiment. For coherence time calculations, we continue to use $\text{FWHM} = 2\Gamma = \frac{2\hbar}{T_2}$, allowing to extract T_2 from the measured FWHM.

General comment 4: Finally, a more general comment: since plasmonic resonators are intrinsically broad (in terms of spectral width), why did the author only focus cryogenic measurements? Their brightening should not degrade significantly at room temperature (in contrast to narrow band dielectric cavities), which would be a very nice achievement.

Our response: Yes, the plasmonic cavity and light enhancement works also well at RT and we currently implement dopant functionalized SWCNTs that show strong RT emission of quantum emitters embedded into our cavities. The work presented in the manuscript under consideration focuses on the particular system of PFO-wrapped SWCNTs that enters the ultra-narrow linewidth regime only at cryogenic temperatures (4K – 120K), above which the linewidth renders broadband (see also our work in ACS Nano 9, 6383 (2015)) and the optical emission intensity strongly quenches. Since narrowband emission is key in our work, including the first demonstration of single-molecule thermometry within this limiting temperature range, we did not attempt to add RT data of other projects.

In conclusion, I think that this work is acceptable for Nature Communications if the previous comments are properly addressed. **Our response:** We thank the reviewer for the detailed response and recommendation for publication of our work in Nature Communications.

Reviewer #2

General comment 1: The manuscript by Luo et al. reports the modification of carbon nanotube photon emission properties by means of coupling to plasmonic bowtie nanocavities. Specifically, the authors find large Purcell enhancements and increased quantum yields as main effects of the plasmonic cavity on the nanotube emission characteristics. Moreover, they speculate that cavity-coupled nanotubes are sources of single indistinguishable photons. Finally, the authors advocate the potential of hybrid nanotube-cavity systems for all-optical thermometry. Having read the manuscript, the supplementary information and the communication within the first two rounds of reviews, I'm hesitant to recommend the paper for publication in Nature Communications in its present form. While the manuscript seems to have improved considerably from its initial version to the revised version under present review, I have to raise additional concerns beyond the critical aspects that have been settled in response to previous reviews.

Our response: We thank the reviewer for the statement that the manuscript has been considerably improved and that all previous critical aspects have been settled in the revised version. We also like to thank the reviewer for raising three additional new concerns aimed to improve our manuscript and the chance to address them in the following.

Comment 1: My main concern, in line with referee #3, is related to the discussion of narrow spectral linewidths and photon indistinguishability. First of all, I didn't find any statistics on the distribution of linewidths for the coupled nanotube-cavity systems at lowest excitation powers. Second, I'm not convinced that the authors can extract the linewidth below the resolution limit of their spectrometer ($\sim 25 \mu\text{eV}$). The section in the supplementary information dedicated to the procedure of zero-phonon-linewidth (ZPL) deconvolution states that sub-resolution Lorentzian contributions to the Voigt profile can be determined provided "good signal to noise ratio". This is clearly not the case in the supplementary Fig. 6. Moreover, the deconvolution is probably strongly compromised by the overall non-Lorentzian lineshape of the nanotube photoluminescence as discussed in section 8. of the supplementary information. In my opinion, the fact that the linewidth of the nanotube in Fig. 5a levels off at low excitation powers is nothing else but a consequence of the spectral resolution limit. This by itself is not troubling if the linewidth had not been used to calculate the two-photon interference visibility shown as inset to Fig. 5a (the set of data in the inset is a compromised projection but is presented in a way as if it has been actually measured). My point of criticism boils down to an unjustified over-interpretation of the experimental data. If the photon indistinguishability is a major result of the paper, the authors should prove it by performing Hong-Ou-Mandel measurements.

Our response: The main issue raised here is that our calculated HOM visibility as an inset to the last Figure has not been additionally confirmed in an experimental way. While we believe there is nothing wrong with the way how we predict a theory value for single photon visibility, as determined from experimental linewidth data that reflect the lower bound on exciton dephasing times as well as lifetime data, we do agree that they are by no means a direct experimental proof that the emitted light is indistinguishable. We have thus decided to entirely remove the Figure and discussions on expected HOM visibility from abstract, discussion, and conclusions and will discuss this in detail elsewhere together with the actual HOM measurements that we currently are setting up.

Regarding the quality of our deconvolution procedure to eliminate the spectrometer response function we like to point out that we indeed have a high enough signal to noise ratio (SNR) reaching up to 14 (12 dB) in Supplementary Fig. 6d and 5 (7 dB) in Supplementary Fig. 6b. This issue was further quantified by Blass (Elsevier 2012) who has shown that at SNR=100 deconvolved values down to 2-2.5 times narrower than

the spectrometer resolution can be reliably extracted. Likewise, to extract linewidth values that are up to 1.5 times (33%) narrower than the spectral resolution limit requires SNR values of about 5-10, which are clearly given in our case with SNR up to 14. We thus have every reason to believe that the standard Voigt deconvolution procedure applied here is indeed valid, with extracted FWHM values of 18-21 μeV for the three lowest pump powers, that are only 10%-14% below the resolution limit of 23.4 μeV .

In addition, the measured Purcell enhanced radiative lifetimes suggest that a significant part of the linewidth is provided via lifetime broadening. In our case the Heisenberg uncertainty limit ($\Delta E \Delta t$) is coinciding near the resolution limit of the spectrometer itself (23.4 μeV). Even in the absence of a spectrometer resolution limit the spectral exciton linewidth is not expected to fall below 17.8 μeV since the measured Purcell enhanced lifetime (37 ps) already dominates the contribution to the ZPL linewidth; $\Delta E = \hbar / 37 \text{ps} = 17.8 \mu\text{eV}$) at the lowest pump power setting where pump induced dephasing is minimized. The effect that the linewidth levels off in Fig.5a is thus the direct manifestation of a Purcell-enhanced emitter reaching its radiative linewidth limit. Note that the comparison to the time-resolved experiment is an independent way to justify that the estimated linewidth values from the deconvolution procedure are scientifically sound.

Regarding linewidth statistics we like to point out that this is provided in Figure 4c showing a histogram of FWHM values of the E_{11} ZPL for 10 reference SWCNTs as well as 21 coupled SWCNTs, for the case of moderate pump power of 0.6 mW. At lowest pump powers (40-200 nW) the ability to resolve linewidth with good SNR ratio is only possible for those SWCNTs that are well coupled, i.e. the best cases, since they have the highest light intensity enhancement. Moderately coupled SWCNTs are thus not measurable reliable at these nW pump powers and thus statistics on linewidth at lowest pump powers cannot be built up and we show our best result in Figure 5.

Comment 2: My second main concern is related to the enhancement of the nanotube absorption. The procedure used by the authors to determine the absorption enhancement is still unclear (see the second report of reviewer #1). On the one hand, the rate equation analysis of a three-level model system as discussed in section 3. of the supplementary information is clearly not applicable as it neglects non-radiative decay. The evolution of the linewidth with the excitation power, on the other hand, is not generic as the ZPL broadening can vary from tube to tube due to different coupling constants to acoustic and optical phonons and different environments of fluctuating charges. Moreover, the connection between motional narrowing effects in semiconductor quantum dots and the power dependence of the ZPL broadening in the present system is highly speculative. Given the uncertainty in the absorption enhancement, the enhancement factor can vary significantly. The authors state that on this basis "one can qualitatively estimate the intensity enhancement" yet they proceed by discussing quantitative numbers.

Our response: We thank the reviewer for pointing out this discrepancy, which we believe is caused by using the Greek letter symbol alpha for two seemingly different things. In SI section 2 we discuss a rate equation analysis with parameter alpha that is related to laser light absorption, but this alpha parameter has nothing to do with the absorption enhancement factor alpha that we discuss in the main text. We have changed the symbol in the revised SI discussion from alpha to delta, which is just a general fitting parameter, while the purpose of this section is to show that measured saturation at highest pump powers is expected in 3-level systems in line with experimental data. This qualitative statement for the saturation regime will not change if the rate equation model fit curves in Fig.S3 would be extended to include another nonradiative channel or not.

Importantly, we note that α characterizing enhanced absorption as used in the main text is determined at a fixed pump power from the difference in linewidth value between coupled and uncoupled system. The linewidth of our reference SWCNTs have excellent reproducibility with an average FWHM of $80 \pm 8.6 \mu\text{eV}$, corresponding to a very small variation of only 11% at pump power of $25 \mu\text{W}$. This is taken as the normalization. In comparison, at the same pump power sent in the linewidth of the exemplary coupled tube is 1.9 times larger. We attribute this broadening to an effective higher pump power in the coupled system causing additional pump-induced dephasing. For this particular tube, a very clear trend following $\Gamma = \sqrt{P}$ is demonstrated, implying that 3.8 times enhanced pump power is needed to reproduce the linewidth broadening. Since the same $25 \mu\text{W}$ for the actual power was sent in also for the coupled case, this power must be effectively better absorbed via the coupling to the nanoantenna, and thus interpreted as 3.8-fold enhanced absorption ($\alpha = 3.8$) causing the observed broadening.

Regarding the concern that the slope relation $\Gamma = \sqrt{P}$ might not hold well for other SWCNTs is well taken. On one hand, the linewidth for coupled tubes does indeed vary from tube to tube due the local difference in coupling strength, as is apparent from the bottom panel in figure 4c ($141 \pm 6 \mu\text{eV}$ to $247 \pm 6 \mu\text{eV}$). Calculating the various α factors in this way for each SWCNT normalized to the $80 \mu\text{eV}$ reference and using $\Gamma = \sqrt{P}$ has contributed to the estimates of the underlying rate enhancement from time-integrated experiments for 21 SWCNTs. We compared these values determined with those from the direct and more trusted time-resolved studies, as is shown by the correlation parameter σ in Figure 4f. The fact that σ is close to unity for all SWCNTs provides evidence that the $\Gamma = \sqrt{P}$ relation holds exceptionally well for all investigated tubes within the errors stated, and thus does at least not vary drastically, thanks to PFO protection and Al_2O_3 passivation.

Lastly, we pointed out a possible connection between motional narrowing effects in semiconductor quantum dots and power dependence of the ZPL broadening. The analogy is plausible considering that excitons in both systems are confined to zero-dimensions, i.e. have quantum-dot like optical properties as is apparent from photon antibunching experiments. We have reformulated this statement in the revised manuscript and point put that the exact microscopic mechanism in PFO-wrapped SWCNTs is still elusive.

Comment 3: Finally, the practicability of the coupled nanotube-cavity system for thermometry has not been evaluated at all. What determines the sensitivity of the system, what is the temperature range for useful implementation, how is the readout compromised by simultaneous heating of the plasmonic cavity, etc?

Our response: We have added the information on the useful temperature range (4K -120 K) of our first demonstration of plasmonic thermometry at the level of individual molecules. The range is limited by the break-up of the acoustic phonon barriers with potential depth around 2 meV (see our Ref.22), above which the exciton linewidth is rendered broadband and no longer reflects temperature. The sensitivity is about ± 5 K, limited by the accuracy of the lineshape analysis to influence the Bose-Einstein occupation factor and thus phonon side-peak intensity considerable. We would like to clarify that in our steady-state experiment there is no compromise by the simultaneous heating of the plasmonic cavity since radiative heating by the plasmonic cavity into its environment is the main source of the temperature that acts back on the SWCNT. We have added this info to the manuscript and removed the statement that our demonstration is practical towards thermometry devices, given that we have not yet demonstrated utility up to room temperature and are limited to 4-120K. It might however become practical if ways are found to engineer polymer backbones that results in stronger phonon localization that would be able to survive up to RT.

General Comment 2: The manuscript, void of the burden of over-interpretation, reports strong Purcell enhancement of a single luminescent (macro)molecule by a plasmonic bowtie nanoantenna. As opposed to molecular dyes with ubiquitous photobleaching, nanotubes with temporally stable emission are particularly viable for the practical realization of such hybrid molecular-nanoplasmonic systems which boost the photon extraction efficiency for emitters with low quantum yields. I find this setting scientifically sound and the successful realization of such a technologically challenging hybrid system remarkable. Placed into an appropriate context, the extensive results of the comparative study of coupled vs. uncoupled nanotubes would justify, in my opinion, publication in Nature Communication without the need of unjustified projections or overstressed potential for applications.

Our response: We again thank the reviewer for the positive assessment and kind words that our work is a remarkable achievement as well as scientifically sound on its key claims. We believe the revised version has removed some of the weakly justified claims, particularly on indistinguishability and practicability of thermometry. We are looking forward to a positive decision.

Reviewers' comments:

Reviewer #1 (Remarks to the Author):

The authors addressed properly my comments and purged their paper from over-interpreted claims. Therefore, I recommend to accept this work for publication in Nat. Com.

Reviewer #2 (Remarks to the Author):

In the revised version of the manuscript the authors have made sufficient changes to all points but one. The remaining point ties to the quantitative limits of the deconvolution procedure used by the authors to determine ultra-narrow linewidths below the resolution limit of their spectrometer which I continue to doubt. I read with interest the argument about the validity of the standard Voigt deconvolution procedure in the presence of finite signal-to-noise ratios. However, the line-shapes in question do not exhibit Voigt profiles but are substantially deformed by phonon sidebands. I'm not sure why the authors insist on determining a linewidth of $18 \pm 3 \mu\text{eV}$ instead of using $23 \mu\text{eV}$ set by the resolution limit that is nearly within the error bars of the (doubtfully) deconvolved value. I would leave the decision on this controversial issue to the editors but wish to insist that the resolution limit of the spectrometer is at least added graphically to Fig. 5a. Also, I noticed that the reference to the original work on photon antibunching in cryogenic carbon nanotube luminescence (Hogele et al, PRL 100, 217401, 2008) is missing and request to include the citation prior to publication. With these two mandatory points for revision I now recommend the publication of the manuscript in Nature Communications.

Response to reviewers

We like to again thank all four reviewers for the many constructive comments that have contributed to significantly improve our manuscript. While three reviewers have accepted our manuscript “as is” the fourth reviewer has pointed out that he/she will recommend the manuscript as well if we add a left-out reference and address one remaining issue related to the validity of the Voigt deconvolution procedure, which we have clarified in the following:

Major comment by reviewer #4: In the revised version of the manuscript the authors have made sufficient changes to all points but one. The remaining point ties to the quantitative limits of the deconvolution procedure used by the authors to determine ultra-narrow linewidths below the resolution limit of their spectrometer which I continue to doubt. I read with interest the argument about the validity of the standard Voigt deconvolution procedure in the presence of finite signal-to-noise ratios. However, **the line-shapes in question do not exhibit Voigt profiles but are substantially deformed by phonon sidebands.** I'm not sure why the authors insist on determining a linewidth of $18 \pm 3 \mu\text{eV}$ instead of using $23 \mu\text{eV}$ set by the resolution limit that is nearly within the error bars of the (doubtfully) deconvolved value. I would leave the decision on this controversial issue to the editors but wish to insist that the resolution limit of the spectrometer is at least added graphically to Fig. 5a.

Our response: We do not believe there is anything controversial here! First of all, the **zero-phonon lineshape (ZPL)** that we fit by the standard Voigt function is **not** affected by any phonon contributions. The acoustic phonon wings that come out at much higher pump powers are separated by about 2 meV away from the ZPL in these PFO-wrapped SWCNTs, as can be seen e.g. in Figure 5b/5c bottom spectrum or in our previous work ACS Nano 9, 6383 (2015) where we provide a detailed lineshape model that includes ZPL and phonon wings. At the low pump powers of 50-200 nW under question in our experiments phonon wings do not emerge significantly out of the noise floor and, importantly, do not contribute spectrally to the ZPL and reside outside of the shown spectral window. The Voigt fit of the zero-phonon line is thus not affected by phonons or “substantially deformed”, which is also clear from the excellent fits shown in Supplementary Figure 6 that are symmetric. Moreover, and as noted previously, our time-resolved experiment shows independently that the lifetime broadening of the ZPL linewidth must be at least $\Delta E = \hbar / 37\text{ps} = 17.8 \mu\text{eV}$ according to the Heisenberg uncertainty relation, which is independent of a spectrometer resolution. These time-resolved experiments thus independently confirm our result from deconvolving time-integrated spectra and provide additional justification that the Purcell-enhanced emitter linewidth saturates against its radiative linewidth limit. We followed the suggestion by the reviewer and have added a dashed line to Figure 5a indicating the $23 \mu\text{eV}$ spectrometer resolution limit.

Minor comment by reviewer #4: Also, I noticed that the reference to the original work on photon antibunching in cryogenic carbon nanotube luminescence (Hogele et al, PRL 100, 217401, 2008) is missing and request to include the citation prior to publication. **With these two mandatory points for revision I now recommend the publication of the manuscript in Nature Communications.**

Our response: We have added the left-out reference by the Imamoglu group (which is indeed a pioneering paper that we regularly cited in all our previous published work on SWCNTs).